# Using Electrical Resistivity Tomography Method to Determine the Inner 3D Geometry and the Main Runoff Directions of the Large Active Landslide of Pie de Cuesta in the Vítor Valley (Peru)

Yasmine Huayllazo [1],*, Rosmery Infa [1], Jorge Soto [1], Krover Lazarte [1], Joseph Huanca [1], Yovana Alvarez [1] and Teresa Teixidó [2]

1   Universidad Nacional de San Agustín de Arequipa, Arequipa 04000, Peru; rinfaa@unsa.edu.pe (R.I.); jsotov@unsa.edu.pe (J.S.); klazarte@unsa.edu.pe (K.L.); jhuancacard@unsa.edu.pe (J.H.); yalvarez.geo@gmail.com (Y.A.)
2   Andalusian Institute of Geophysics of Granada University, 18071 Granada, Spain; tteixido@ugr.es
*   Correspondence: yhuayllazo@unsa.edu.pe

**Abstract:** Pie de Cuesta is a large landslide with a planar area of 1 km$^2$ located in the Vítor district, in the Arequipa department (Peru), and constitutes an active phenomenon. It belongs to the rotational/translational type, which concerns cases that are very susceptible to reactivation because any change in the water content or removal of the lower part can lead to a new instability. In this context, a previous geological study has been decisive in recognizing the lithologies present and understanding their behavior when they are saturated. But it is also necessary to know the inner "landslide geometry" in order to gusset a geotechnical diagnosis. The present study shows how the deep electrical profiles (ERT, electrical resistivity tomography method), supported by two Vp seismic refraction tomography lines (SVP), have been used to create a 3D cognitive model that would allow the identification of the inner landslide structure: the 3D rupture surface, the volume of the sliding mass infiltration sectors or fractures, and the preferred runoff directions. Moreover, on large landsides, placing the geophysical profiles is a crucial aspect because it greatly depends on the accessibility of the area and the availability of the physical space required. In our case, we need to extend profiles up to 1100 m long in order to obtain data at greater depths since this landslide is approximately 200 m tall. Based on the geophysical results and geologic information, the 3D final model of the inner structure of this landslide is presented. Additionally, the main runoff water directions and the volume of 90.5 Hm$^3$ of the sliding mass are also estimated.

**Keywords:** large landslide; electrical resistivity tomography (2D-ERT); Vp-seismic refraction tomography (2D-SVP)

## 1. Introduction

Landslides are geological phenomena that consist of downhill earth movements influenced by gravity and caused by natural and anthropogenic factors such as rainfall episodes, agricultural activities, and civil land activities, among others [1–3]. These complex phenomena have a large socioeconomic impact and require multidisciplinary geosciences research to determine their internal structure and surrounding environment to facilitate stability analysis and risk mitigation [4–6].

From a geological standpoint, landslide occurrence is frequently related to shear strength reduction and fluidization processes suffered by clayey soils when water flux occurs through the subsurface [7–11]. To understand the occurrence and evolution of a landslide, it is of prime importance to obtain information about the local geology, subsurface hydrogeological conditions, and depth of the failure surface. Frequently, such information

cannot be easily obtained because it is needed to drill expensive boreholes over the landslide zone. In this context, geophysical methods are increasingly being used with great success to define the physical properties of soils [12,13] and to obtain geologic and hydrogeological information about the study area [14,15]. Nowadays, landslides are also monitored in real time using different remote sensors to predict their evolution [16,17].

The use of geophysical methods to study landslides presents several advantages: (i) they are non-destructive techniques, (ii) they can reach the maximum target depth of interest as an extensive mode, and (iii) the resulting models may show the inner geometry of landslide mass, even in cases of structural complexity. Among the set of prospecting methods available, the 2D Electrical Resistivity Method (2D-ERT) is very useful in the study of landslides because it is sensitive to variations in water content and the grain size of materials, providing detailed subsurface models [18,19]. Meanwhile, Seismic Vp Velocity (2D-SVP) is a useful surface method for obtaining information on the different compactions of materials to establish stability studies of the sliding mass. And in cases such as this, where the flow water is salinized, this allows discrimination at the top of the basal layer [19]. However, this method is used less in landslide studies [20–22].

Peru, due to its geographical location and geomorphological conditions, is a country that is exposed to several natural phenomena, where mass movements are one of the most common events that cause human and economic damage. One of the largest and most destructive landslides in the Vítor Valley (Arequipa) is the Pie de la Cuesta landslide, which currently has a main scarp that is 1 km long and a planar area of approximately 1 km$^2$. This mass movement has been taking place since 6 October 1974 [23], having produced a large-scale collapse on 9 January 1975 that buried the town of Pie de Cuesta and affected more than 12 families who had to be relocated [24]. In 2018, the Perú National Research Program funded the National University of San Agustín de Arequipa to carry out a research project (IBAIB-03-2018-UNSA) to characterize this landslide.

For this purpose, a geophysical survey consisting of electrical and seismic profiles was carried out on the landslide to obtain the main inner geometry of the sliding layers with these non-destructive methods. This paper summarizes the study carried out to correlate electrical (2D-ERT) and seismic (2D-SVP) profiles with geological information and to obtain the internal structure of a large landslide and its main water flow directions.

## 2. Study Area

### 2.1. Geological Description of Study Area

The Pie de Cuesta landslide is located at the foothills of agricultural terrain corresponding to the La Joya Antigua system (Figure 1), which is a plain of more than 3500 ha of crops where the irrigation method is mostly flooding. So, the accumulation of groundwater is caused by irrigation. Geologically [24], the study area is formed by three formations (Fm) (Figure 1): At the bottom, the Lower Moquegua Fm (96 m thick) is mainly composed of polymathic conglomerates of rounded and surrounded clasts with a sandy matrix and intercalated with arkosic sandstones and shales. In the middle, the Upper Moquegua Fm (100 m thick) is placed. This formation is composed of red shales alternated with gypsum layers, whose upper part contains red shales with layers of thick tuffaceous sandstones. The top formation is the Millo Fm (55 m thick), which is mainly composed of conglomerates of rounded clasts of intrusive and volcanic origin with a sandy matrix intercalated with silty sandy gravels; a layer of cream-colored ignimbrite is also observed. In addition to these formations, the study area also presents alluvial, coluvial and fluvial materials.

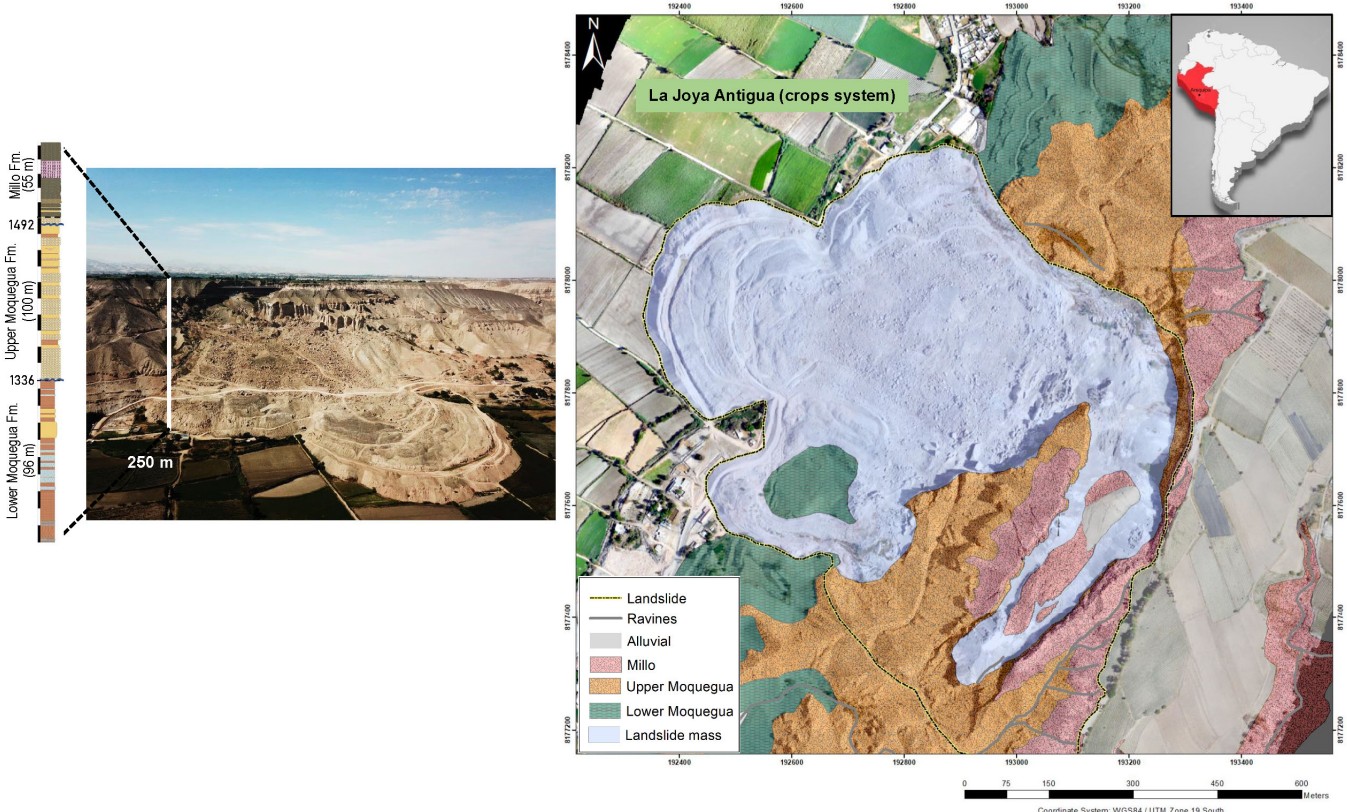

**Figure 1.** Geological formations of the Pie de Cuesta landslide. On the left, the stratigraphic column has a 250 m thickness. On the right, the geological outcrops over the ortofoto-map. In the northwest are the irrigated crops of the Joya Antigua, which act as a source of landslide activation.

The croplands are placed over the Millo Fm, which behaves as a permeable unit, allowing the filtration of irrigation water due to its composition and poor consolidation. In the main wall of the landslide, it can be seen that the water seeps into the Upper Moquegua Fm through fractures and cracks. Two important water tables can also be distinguished: the first one is in the Millo Fm and Lower Moquegua Fm contact (1492 m s.l.), and the second is in the Upper and Lower Moquegua Fms contact (1336 m s.l.), resulting in an interface of high humidity.

If we consider the previous geological description, this means that a 250 m thick column of geological materials is involved in this landslide. This constraint implies that the lengths of the geophysical profiles must be long enough to guarantee a subsurface depth of inspection below the second water table: about 200 m in depth.

*2.2. Description of the Landslide through Historical Aerial Imagery*

The Cuesta landslide began in 1975 and remained active until the 1990s [25]. In 2016, it reactivated, which has continued up to the date of this study [26]. Figure 2 shows the historical evolution of this landslide. The first image corresponds to a 1945 ortho-photomap showing the state of the terrain before the catastrophic failure occurred in 1975 [27,28]. The second image is from a 2015 Google Earth photo, prior to the 2016 reactivation. And the third image is an orthomosaic from 2019, obtained from photogrammetric processing of a drone flight. In the first map of 1945, the landslide boundaries of the years 2015 and 2019 have been marked, showing the characteristics of the landslide advance: between 1975 and 2015, the major collapse occurs, sliding the materials toward the SE margin in the direction of the valley, while the reactivation preference is for 2019 and especially of the main lobe that is close to the right flank of the landslide.

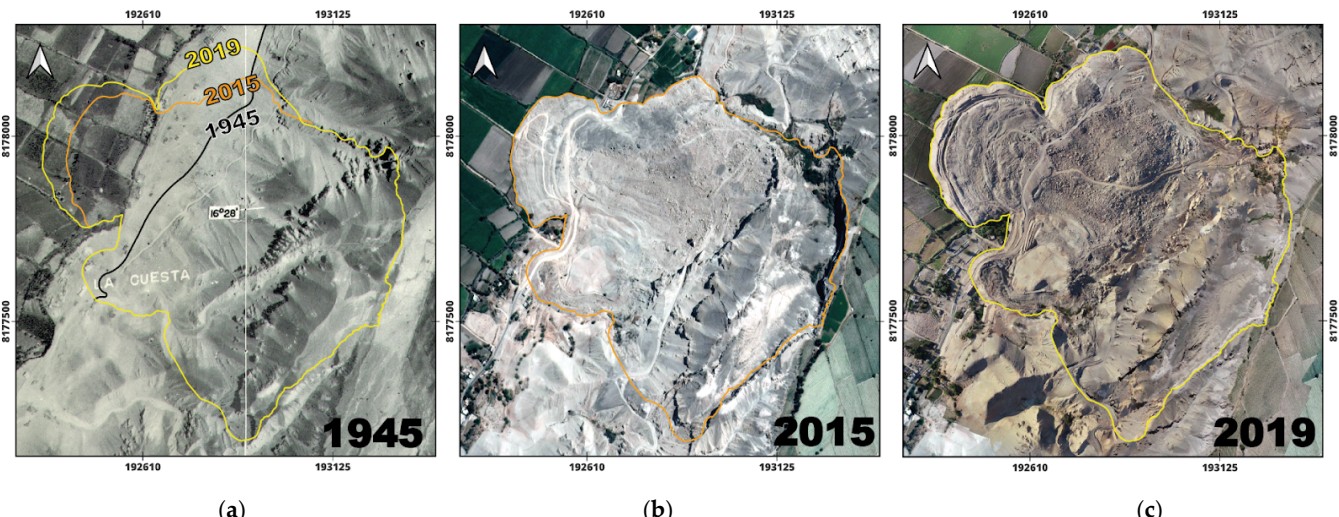

**Figure 2.** Pie de la Cuesta landslide historical evolution. (**a**) Ortho-photomap of 1945 showing the primitive slope limits (black line), the landslide boundaries relative to the first stage at 2015 (orange line), and the limits of the last reactivation in 2019 (yellow line). (**b**) Google Earth image corresponding to 2015. (**c**) Orthomosaic of 2019, obtained from aerial photogrammetry using a drone.

## 3. Materials and Methods

Figure 3 shows the locations of geophysical surveys conducted in this study, where the main method applied was the electrical resistivity tomography profiles (2D-ERT), as they provide suitable electrical resistivity models that are sensitive to lithological changes and the water content of the materials. And it is useful in complex geological conditions [29,30]. As a complementary method, two velocity P-wave seismic refraction tomography surveys (2D-SVP) were carried out in order to evaluate the safety of the main materials present in the landslide [31,32].

### 3.1. Geophysical Non-Invasive Data Acquisition

The purpose of electrical surveys is to determine the subsurface resistivity distribution by making measurements on the ground surface. The electrical c.c. surveys were acquired by a set of electrodes placed on the ground at defined distances, as shown in Table 1. In general terms, the method consists of injecting a current into the ground using two current electrodes and measuring the resulting voltage difference of the generated electric field at two other electrode pairs of the section (potential electrodes). With the Ohm law and the inversion procedure, the result is a parametric 2D model with ground-apparent resistivity distributions. As seen in Figure 3c, the electrical dataset was acquired using an Elect Pro-10 resistivimeter (Iris Instruments, Inc.; Orleans, France) which is a 10-channel receiver especially designed to record deep profiles. In this study, the resistivimeter was used in conjunction with the Elec Pro Switch, which allows 48 electrodes to be connected to the device by multi-electrode cables. The receiver pulse signal was 2 s, and the maximum input voltage was 15 V (automatic gain). For the measurements, we used a pole-dipole electrode array configuration, which has good horizontal coverage, although it has a higher signal compared to others electrode configurations. It is suitable for this deep work because it is not sensitive to telluric noise [33]. The main technical features of this data acquisition are summarized in Table 1.

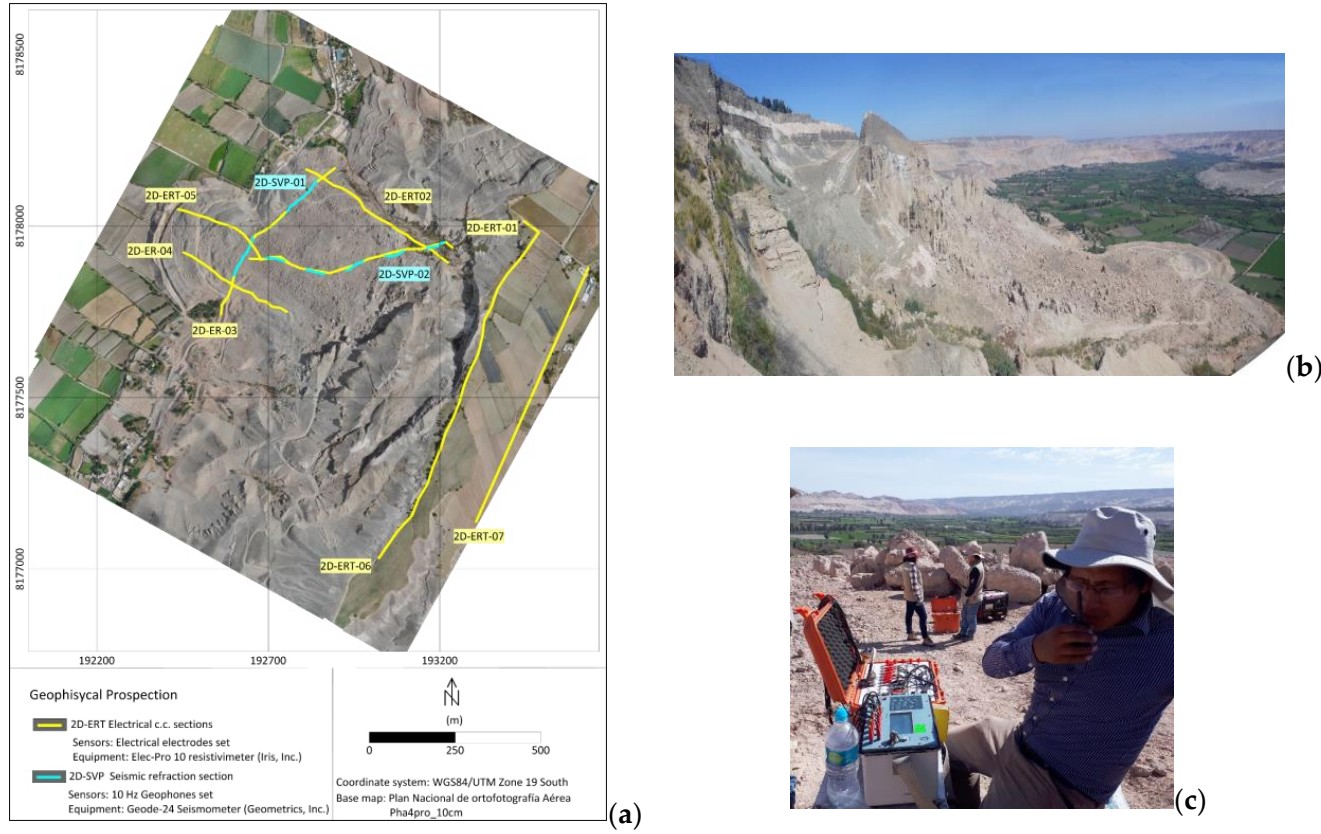

**Figure 3.** (**a**) Locations of geophysical surveys in the study zone. The 2D-ERT profiles are marked with yellow lines, and the 2D-SVP sections are marked with blue lines. Note that the 2D-SVP matches the 2D-ERT-01 and 3D-ERT-03 profiles. (**b**) The absence of sections in the southeast sector of the landslide is due to the rugged topography; see image. (**c**) The electrical equipment is especially designed to reach the required depth, mainly consisting of an Elect Pro-10 resistivimeter (blue box in the foreground), a channel switch-Pro (orange box in the background), and a current injector device (at the back).-Iris Instruments, Inc. (2023) from www.iris-instruments.com/elrec-pro.html (accessed on 29 October 2023).

**Table 1.** Main technical features of the geophysical equipment and data acquisition.

|  | Profiles | Sensors Spacing | # of Sensors | Sensor Array | Total Length | Reached Depth |
|---|---|---|---|---|---|---|
| ERT survey | 2D-ERT01 | Electrodes at 40 m | 33 | Pole-Dipole | 640 m | 160 m |
|  | 2D-ERT02 | Electrodes at 30 m | 35 | Pole-Dipole | 510 m | 120 m |
|  | 2D-ERT03 | Electrodes at 25 m | 45 | Pole-Dipole | 550 m | 140 m |
|  | 2D-ERT04 | Electrodes at 25 m | 29 | Pole-Dipole | 350 m | 90 m |
|  | 2D-ERT05 | Electrodes at 30 m | 27 | Pole-Dipole | 390 m | 100 m |
|  | 2D-ERT06 | Electrodes at 50 m | 45 | Pole-Dipole | 1100 m | 220 m |
|  | 2D-ERT07 | Electrodes at 50 m | 35 | Pole-Dipole | 850 m | 200 m |
| SVP survey | 2D-SVP01 | Geophone at 5 m | $2 \times 24$ of 20 Hz | 13 shots | 420 m | 240 m |
|  | 2D-SVP02 | Geophone at 5 m | $2 \times 24$ of 20 Hz | 19 shots | 600 m | 70 m |

For the seismic method, the Vp refraction tomography considers the first arrival times of the P-waves produced by a controlled source at different sensor stations (geophones). In this study, the seismic 2D-SRT profile was acquired with a Geode system (Geometrics Inc.; San Jose, USA) that controls 24 vertical geophones of 20 Hz of natural frequency. We used custom-made low-energy explosives as the seismic source in order to achieve a good

signal at far offsets. The result was a 2D seismic model of the subsurface showing the Vp velocity distribution [34].

The spread of the seismic recording data (Table 1) was designed in order to ensure ray coverage along the entire section. Accordingly, the center shot positions were repeated to link the recording units with 24 channels each. The sample rate was 0.125 ms, and the total recording time was 0.5 s. The seismic sources were placed at 0.5 m depth.

### 3.2. Electrical Data Processing

The ERT method calculates the electrical resistivity properties of rocks beneath the surface and is a well-established method in near-surface characterization studies [34]. In this study, to calculate the resistivity models from the apparent field-measured parameters, the Res2dinv commercial software (V5.0, Bentley Systems; Exton, PA, USA) was used. This code is designed to obtain 2D (and 3D) geoelectric models by applying the inversion computational technique [35,36]. Due to the fact that the subsurface resistivity is strongly influenced by a rock's properties such as porosity, mineral composition, fluid content, and fault structure [37,38], different processing options have been tested for the inversion procedure. In particular, we wished to determine the influence of the finite mesh grid size and the effect of the damping inversion factors. The damping factor leads to a stabilization of the solution but produces a smoothed resistivity model [37]. In our case, a smoothed section was not desirable because a large landslide is characterized by a heterogeneous underground. Consequently, we selected medium damping factors to stabilize the calculation with a maximum mesh refinement of the parameter models. Other aspects included the choice of the inversion algorithm. When the subsurface has vertical discontinuities, such as falls, the conventional least squares smoothness-constrain method tends to smear the boundaries, and it is best to operate with the robust constrained inversion method, which is less sensitive to resistivity contrasts but gives a high apparent resistivity, although with more fitting errors [38]. Finally, we opted for the following inversion parameters: a refinement cell model defined by four nodes for electrode spacing, a medium inversion damping factor (0.15), and a robust constrained method. The convergence iterations and the absolute error between the measured and calculated apparent resistivities are summarized in Table 2. Figure 4 is an example of the inversion processing for the 2D-ERT-IP-02 profile.

**Table 2.** Inversion parameters of the geophysical models.

|  | **2D -Sections** | **# of Iterations** | **Abs. Error** | **Reached Depth (m)** |
|---|---|---|---|---|
|  | 2D-ERT01 | 8 | 8.9 | 167 |
|  | 2D-ERT02 | 7 | 8.6 | 120 |
|  | 2D-ERT03 | 9 | 6 | 125 |
| ERT survey | 2D-ERT04 | 10 | 8.8 | 95 |
|  | 2D-ERT05 | 9 | 9.8 | 126 |
|  | 2D-ERT06 | 10 | 12.3 | 215 |
|  | 2D-ERT07 | 10 | 19.8 | 210 |

In this study, the 2D-ERT-IP-06 and ERT-IP-07 electrical profiles presented more errors; paradoxically, these two surveys are located outside the landslide behind the northern scar, where a horizontal stratification is assumed (Moquegua Fm). As it will be seen, the cause is due to a fault system detected in this setting.

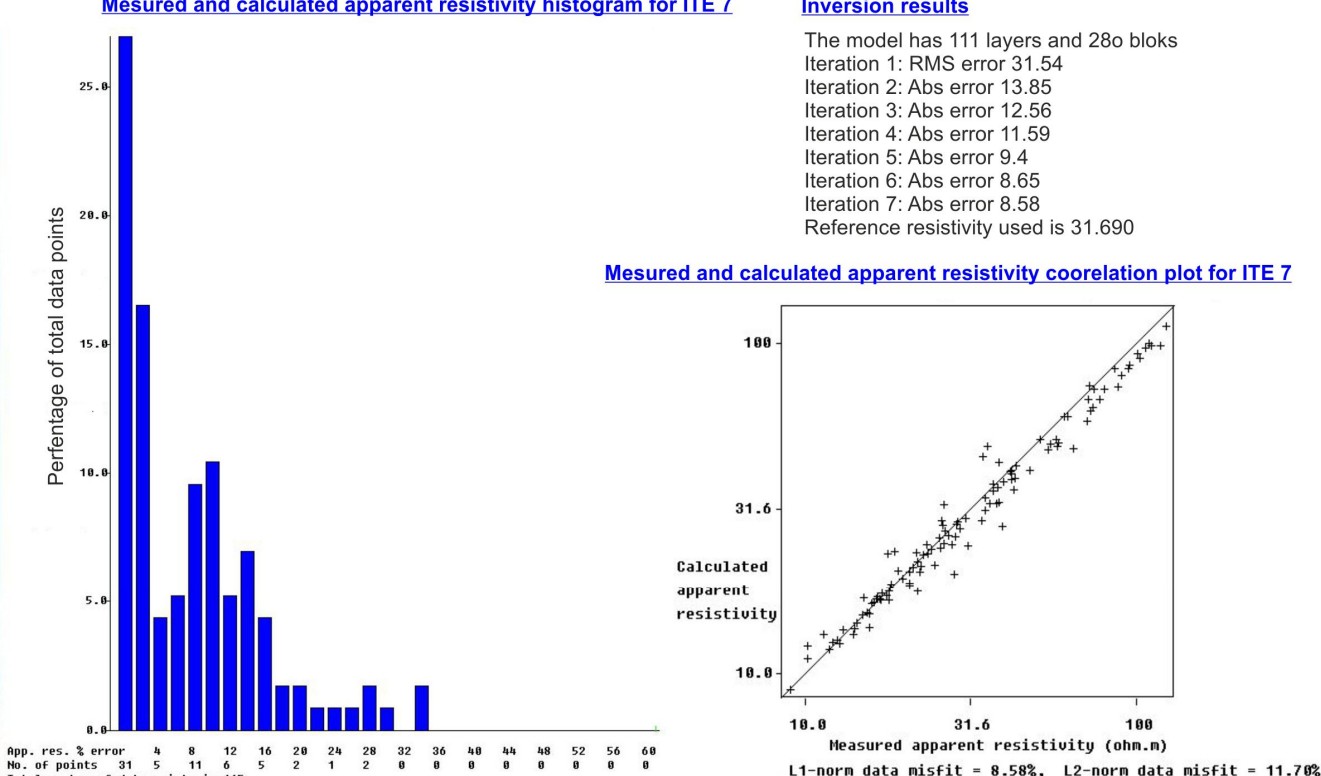

**Figure 4.** General information resulting from the inversion processing flow for the 2D-ERT-02 profile. The histogram shows the absolute error (%) between calculated and measured apparent resistivities in the 7th iteration. The upper text ("Inversion results") indicates the convergence in successive iterations. The second graph is the correlation plot between measured and calculated apparent resistivities in the 7th iteration.

### 3.3. Seismic Refraction Data Processing

P-wave travel time tomography is currently a well-established and broadly used inversion scheme to resolve Vp velocity structure [39]. Travel time tomography is a non-linear problem in geophysics because the deflection of seismic rays depends on the unknown velocity of the subsurface structure. Common approaches use inversion algorithms consisting of picking the first arrival travel times of the P-waves and searching for the most plausible velocity model that can reproduce the observables by minimizing the time difference between the estimated travel times. Theoretical travel times are thus calculated using a ray-tracing forward modeling scheme.

In our study, the first arrivals were handpicked from the shot records (Figure 5a), and we used a commercial Rayfract code (Intelligent Resources Inc. Software, Winnersh, UK) for the inversion proposal (Table 3). To determine the subsurface velocity distribution from the first arrivals picked travel times, we used the Delta-t-v inversion method [40]. This technique is based on the common mid-point (CMP) refraction concept, which considers the CMP travel times as a function of the independent variable CMPx coordinates and the CMP constant offsets (Figure 5c). It starts by determining the velocity at the base of a layer from CMP travel time curves (Figure 5b), and then it numerically inverts the velocity at the top of the gradient layer. The algorithm automatically identifies precise time delays on CMP curves, transforming these delays into velocity–depth anomalies. A 1-D Vp velocity–depth function is constrained beneath each CMP. All 1D velocity–depth functions are integrated through a gridding scheme, building up a final 1.5D velocity model. A simple and smooth 1D velocity model is needed to initialize the process. This is obtained by laterally extending a simple 1D layered model along the profile.

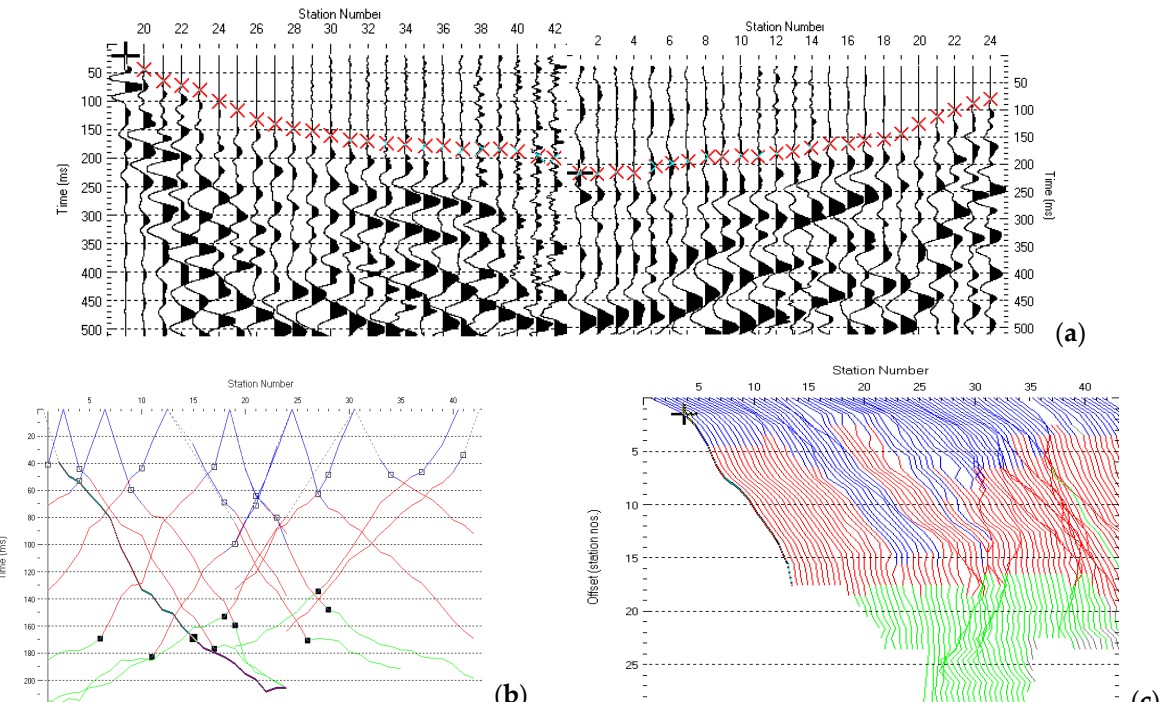

**Figure 5.** Basic inputs to resolve Vp subsurface distribution. (**a**) Examples of the first time picked for the P-waves in two shots viewer (red crosses). (**b**) Space–time graphs of all picked first arrivals. (**c**) The first time picks as a function of the independent variables CMPx (station) and the CMP constant offset; curves are presented with a velocity reduction of 4500 m/s.

**Table 3.** Seismic fitting between modeled and picked travel times. Normalized RMS error is the RMS error, divided by average pick time of all traces modeled.

|  | Profile | Normalized RMS Error | # Traces Modeled | # of Iterations |
|---|---|---|---|---|
| P-wave seismic survey | 2D-SVP01 | 8.5 (%) | 279 | 8 |
|  | 2D-SVP02 | 9.1 (%) | 281 | 7 |

Figure 5c shows the first arrival times in the CMP scheme. In this way, the effects of dipping layers are averaged and minimized. The travel times are smoothed by stacking CMP-sorted travel time curves over 40 adjacent CMPs. Subsequently, each curve is "Deltat-v inverted". Table 3 summarizes the quantitative indicators in the Vp final models, which suggests that they have a relatively high degree of reliability.

## 4. Results and Interpretation

Landslides are considered a natural process that occur in a variety of geologic settings, either as soil mass movement, debris flow, rockfall, or combinations of both. In this case, the main triggering factor has been the continuous agricultural irrigation of the crop fields located to the north (La Joya Antigua, Figure 1). So, for years (around 1968 until today), irrigation water has been infiltrating the ground through surface cracks and sink holes to saturate the subsurface. In fact, irrigation not only triggers landslides in this area, but the occurrence of a landslide itself also increases the chances for new events to happen, as observed in the reactivation of 2016 (Figure 2).

### 4.1. Basis for the Interpretation

Landslides are classified by their type of movement [41,42], and actually the scheme terminology is suggested by the UNESCO Working Party on the 'World Landslide Inven-

tory' (WP WLI 1990, WP/WLI 1993). The four main types of movement are caused by the following factors:

(1) Falls. These landslides involve the collapse of materials from a cliff or steep slope. Falls usually involve a mixture of free falls through the air, either bouncing or rolling. A fall-type landslide results in the collection of rock or debris near the base of a slope.

(2) Topples. Topple failures involve the forward rotation and movement of a mass of rock, earth, or debris off a slope. This kind of slope failure generally occurs around an axis (or point) at or near the base of the block of rock.

(3) Flows. Flows are landslides that involve the movement of material down a slope in the form of a fluid. Flows often leave behind a distinctive, upside-down funnel-shaped deposit where the landslide material has stopped moving. There are different types of flows: mud, debris, and rock (rock avalanches).

(4) Rotational and translational slides. Rotational slides occur on curved slip surfaces where the upper surface of the displaced material may tilt backward toward the scarp, whereas a translational (or planar) landslide is a downslope movement of material that occurs along a distinctive planar surface of weakness, such as a fault, joint, or bedding plane. Some of the largest and most damaging landslides on Earth are translational. These landslides occur at all scales and are not self-stabilizing. They can be very rapid when discontinuities are steep.

Figure 6 contains general schemes of the internal structure of a rotational/transitional slide and the usual nomenclature of its main parts. From these drawings, it can be inferred that the main goal of this study is to determine the inner geometry–geology. So, this means that the application of geophysical methods is recommended to develop structural and hydrogeological inner models for this sliding mass.

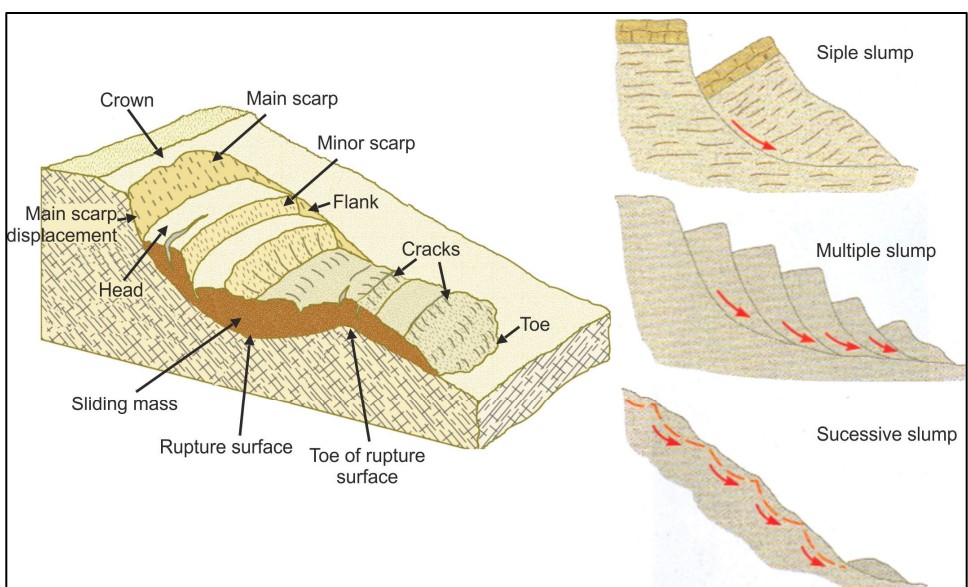

**Figure 6.** Left image shows the inner geometry of the rotational/translational landslide and its nomenclature. Images on the right represent the different degrees of complexity that slippage can have. Modified from Varnes 1996 [34].

Taking this classification into account, the Pie de Cuesta landslide belongs to this type. Such cases are often very susceptible to reactivation because, after the slip has occurred, the equilibrium position is reached when the torque decreases. So, any change in the water content or the removal of the lower part can lead to a new instability and thus, a reactivation of the slip. Therefore, a diagnosis of the inner geometry of these phenomena is needed to make safety estimations.

For this purpose, the 2D-ERT resulting models are interpreted according to this scheme and the hydrogeological information [14,43,44], while the 2D-SVP models focus on detecting the compaction of these sedimentary materials, particularly bedrock geometry. Figure 7 includes the correlation between the geologic materials (lithologic column), the resistivity ranges (Ωm), and the P-wave velocities (m/s). Given that, low- resistivity values correspond to saturated materials due to the high mineralization (salinized) of irrigation water. Hydrological studies carried out in the area indicate water conductivity between 5 and 8 mS/cm equivalent to 1.25–2 Ωm [32]. On the other hand, low velocities are related to soft materials, whereas an increasing velocity is proportional to their compaction.

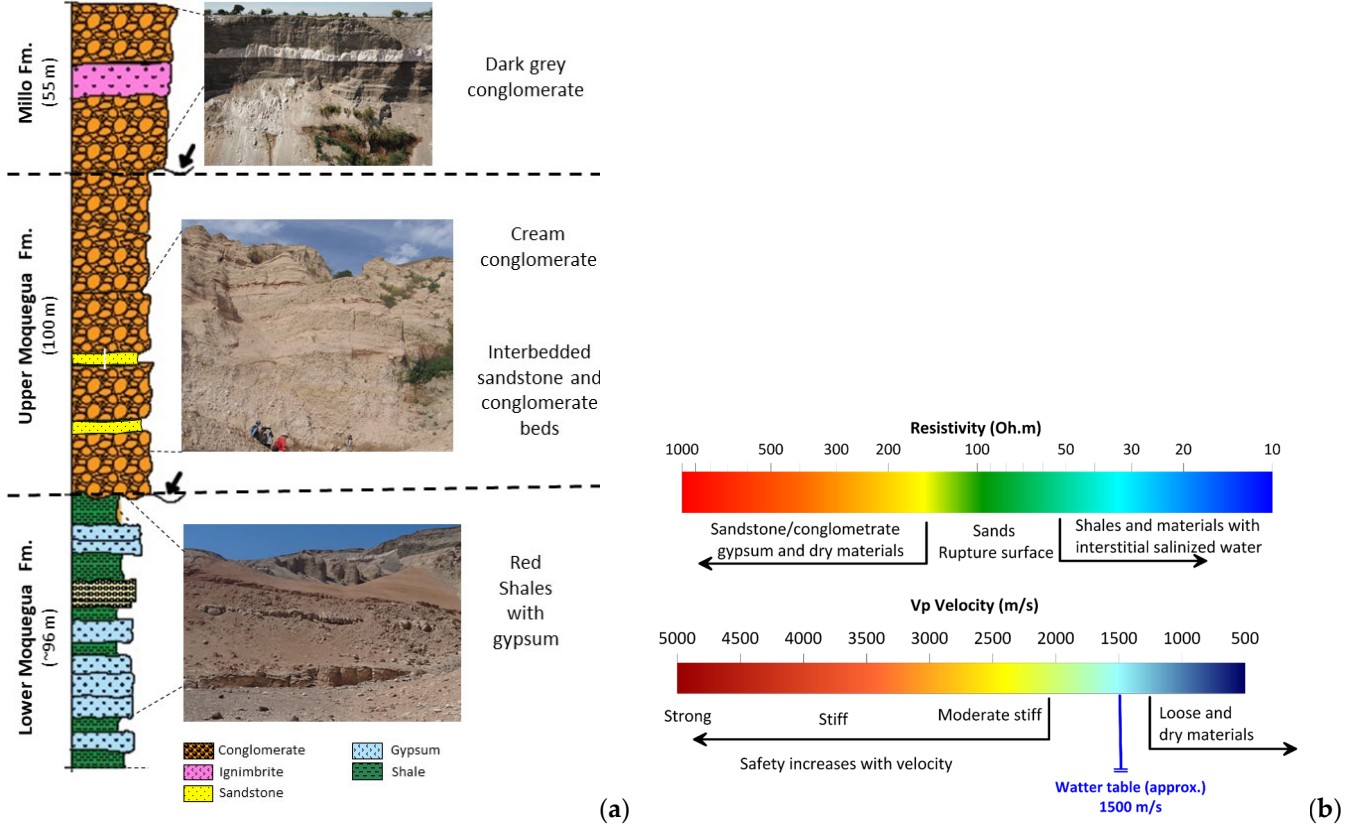

**Figure 7.** (**a**) Representative lithologic column of the study area. (**b**) Correlation between the geophysical parameters and the lithology, hydrology, and compaction of the materials present in this landslide.

On the velocity scale, 1500 m/s has been marked as the approximate level for the water table. This value corresponds to the velocity of P-waves through the water (under standard conditions) and in free or multilayer aquifers. As in the case of this landslide, this value indicates the level at which the unconsolidated sediments are saturated [32].

### 4.2. Interpretation of the 2D-ERT Profiles

The electrical profiles were interpreted according to the two geological targets where they were acquired. The ERT-06 and ERT-07 profiles are placed in the non-sliding area, behind the scarp. And the other five profiles (from ERT-01 to ERT-05) are located within the landslide (Figure 3).

Concretely, the first two profiles are located in the crop fields. ERT-06 is the closest to the escarp, and ERT-07 is parallel and 200 m furthest away. As the sediments in these profiles are not sliding, it was possible to establish a stratigraphic correlation, and by fixing the boundaries of three geological formations, the infiltration impact of irrigation water on the materials was assessed. In both profiles (Figure 8a), the two water tables at 1492 m s.l.

(Millo Fm and Lower Moquegua Fm contact), and 1336 m s.l. (Upper and Lower Moquegua Fms contact) were marketed. With these divisions, it can be seen that there are infiltration areas in the crop fields. The highest infiltration rate occurs in the northern area, coinciding with the side of the steepest scarp, while toward the south, this infiltration decreases at the ends (profile ERT- 07) at 950 m long, where the crown finishes.

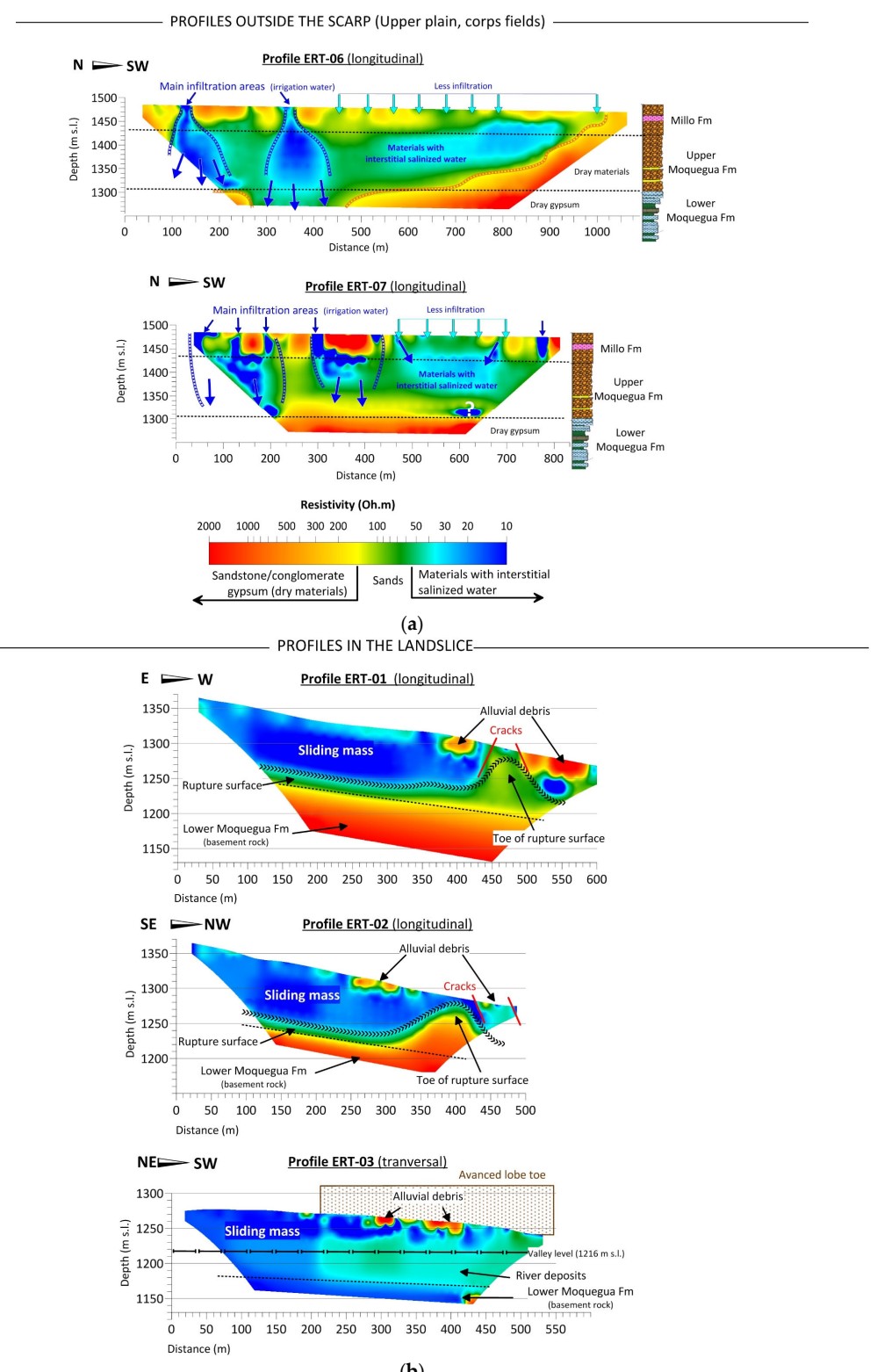

**Figure 8.** *Cont*.

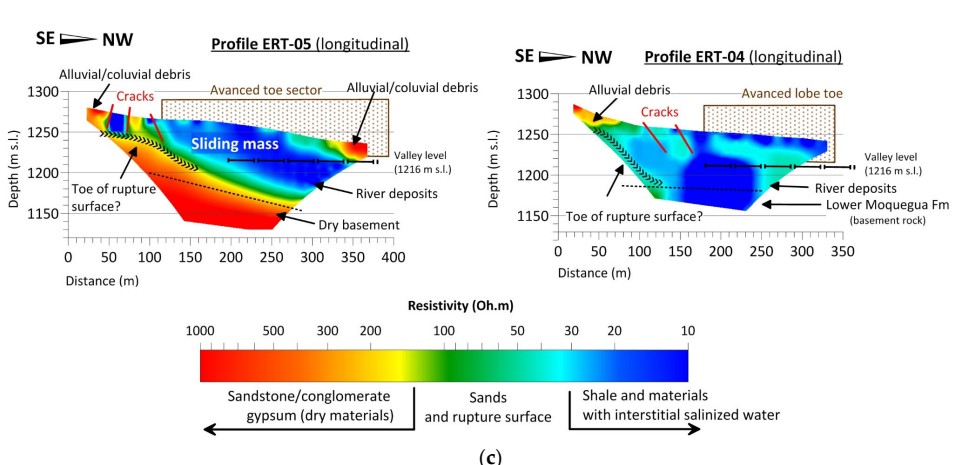

**Figure 8.** (**a**) Interpretation of the electric profiles placed outside the landslide (ERT-06 and ERT-07, on previous page). (**b**,**c**) Interpretation of the electric profiles placed within the landslide (from ERT-01 to ERT-05).

In general, the conglomerate is a permeable lithology that has a high resistivity between 200 and 800 $\Omega$m (depending on the amount of interstitial water and its composition). In this case, the low resistivities detected in the Upper Moquegua Fm indicate that these sediments have saline water and resistivity values lower than 10–20 $\Omega$m (blue), which suggests the existence of saturated sectors. The result is an interface of high humidity that can reactivate the landslide if any instability occurs. This is in accordance with the 2015–2019 years of reactivation, where the crown was mostly displaced in the most conductive zones (Figure 2). Another interesting aspect is the plume shape of the two main infiltration zones; in both cases, they appear to be flowing toward the Lower Moquegua Fm contact. If we look at profile ERT-06, the section between 260 and 460 m is the most affected for water plume, and considering the proximity to the crown, this segment could be the trigger for another future reactivation.

The profiles within the landslide show different aspects of the internal structure of the landslide depending on where it was made (Figure 8b). In this way, the ERT-01 and ERT-02 profiles are placed in the upper part, below the main scarp (Figure 2). They are nearly parallel and run longitudinally to the slipped materials. In them, we can see a sliding mass characterized by low resistivities (lower than 10–20 $\Omega$m, blue values) bounded by the slip plane (rupture surface), which defines the top of the dry basement with high resistivities (higher than 200 $\Omega$m, yellow-red values). At the end of these two profiles, we can see the elevation corresponding to the toe of the rupture surface, where the debris (remains dray) accumulates and a system of cracks is detected on both sides of this elevation. This relief is distinctive for the rotational/transitional landslide, after which the fluidization zone begins. The ERT-01 profile has lower resistivities for the toe rupture surface (between 60 and 160 $\Omega$m, yellow-green values), suggesting that the materials may have a higher mineralized water content. It is precisely this profile that is located in the most advanced lobe of the slide.

Longitudinal profiles ERT-04 and ERT-05 (Figure 8c) are placed after the toe of the rupture surface, and they show the sliding mass in the fluidization zone, where the detected resistivities are slightly higher (around 30 $\Omega$m, light blue values). This aspect can be interpreted in two ways: (i) as a loss of water content of the materials because they have come into contact with river deposits; (ii) as a mixture with the fresh water of the valley.

Finally, profile ERT-03 (Figure 8b) was acquired transversally behind the toe rupture surface in the fluidization zone. In this case, it is observed that the sliding mass is more resistive in the central lobe, coinciding with the two previous profiles, while at the NE,

under the second advancing lobe, it is more conductive. This could suggest a greater plasticity of materials, which would give this lobe a higher risk of advancing.

### 4.3. Interpretation of the 2D-SVP Seismic Profiles

The SVP-01 seismic profile (Figure 3) is planned almost coincidentally with the transversal electrical profile ERT-03 and SVP-02 with the longitudinal ERT-01 profile. In Figure 9, both electrical and seismic equivalent models have been jointly interpreted. In the SVP-01 model, we can see three velocity layers (Figure 9a). The first one is a thick layer (200 m) made up of loose materials with a low velocity (Vp < 1400 m/s) that can be associated with the sliding mass detected in the ERT-03 profile. Below this, an intermediate layer has been detected related to moderately stiff materials comprising between 1500 m/s and 3000 m/s. These values have been set in ERT-03 like respective iso-velocity lines (white and red lines in Figure 9a), and we can infer that the 1500 m/s value marks the water level (approx.), while 3000 m/s defines the top of the strong materials linked with the Lower Moquegua Fm. On the NE side of both models (SRT-01 and ERT-03), this middle layer can be correlated with the sliding mass, while at the SW end, it seems to coincide with the river deposits (materials of the valley). Note that the contact between these two geological environments is described by a local relief (200 m) that could act as a "barrier". In Figure 10b, it can be seen how this relief coincides with the end of the toe subsurface.

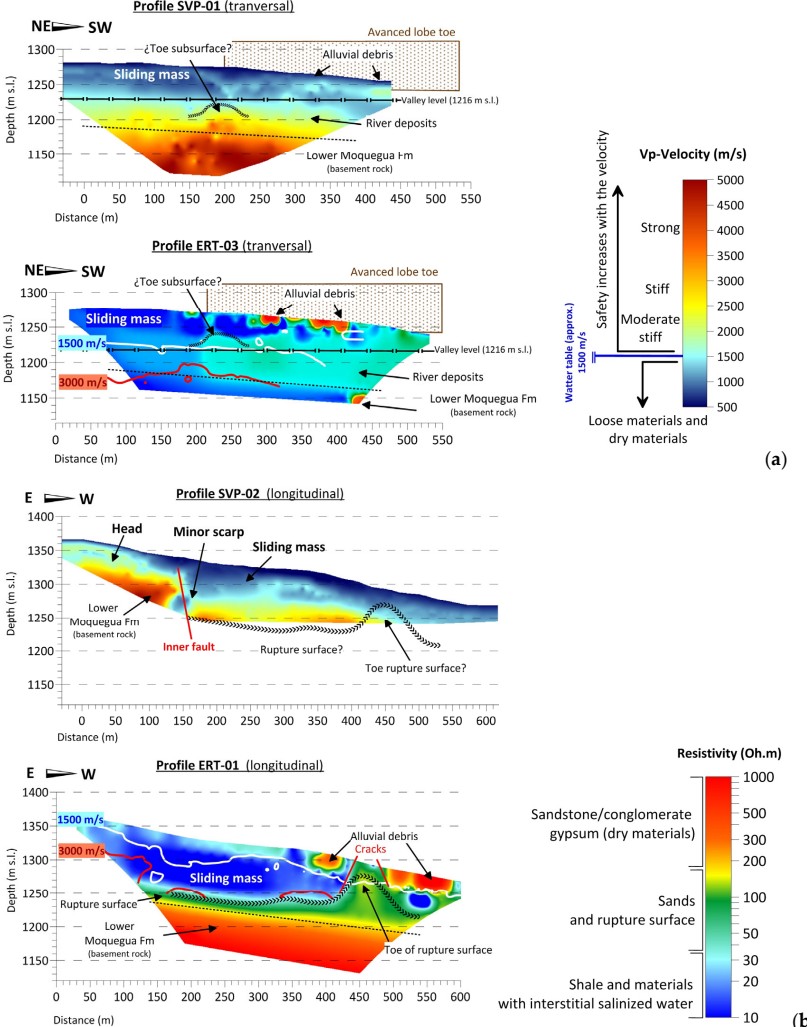

**Figure 9.** Interpretation of the seismic profiles placed within the landslide. (**a**) The SVP-01 is almost coincident with the ERT-03 profile and their interpretation has been done by correlating seismic and electrical responses. (**b**) The same procedure has been used for the SVP02.

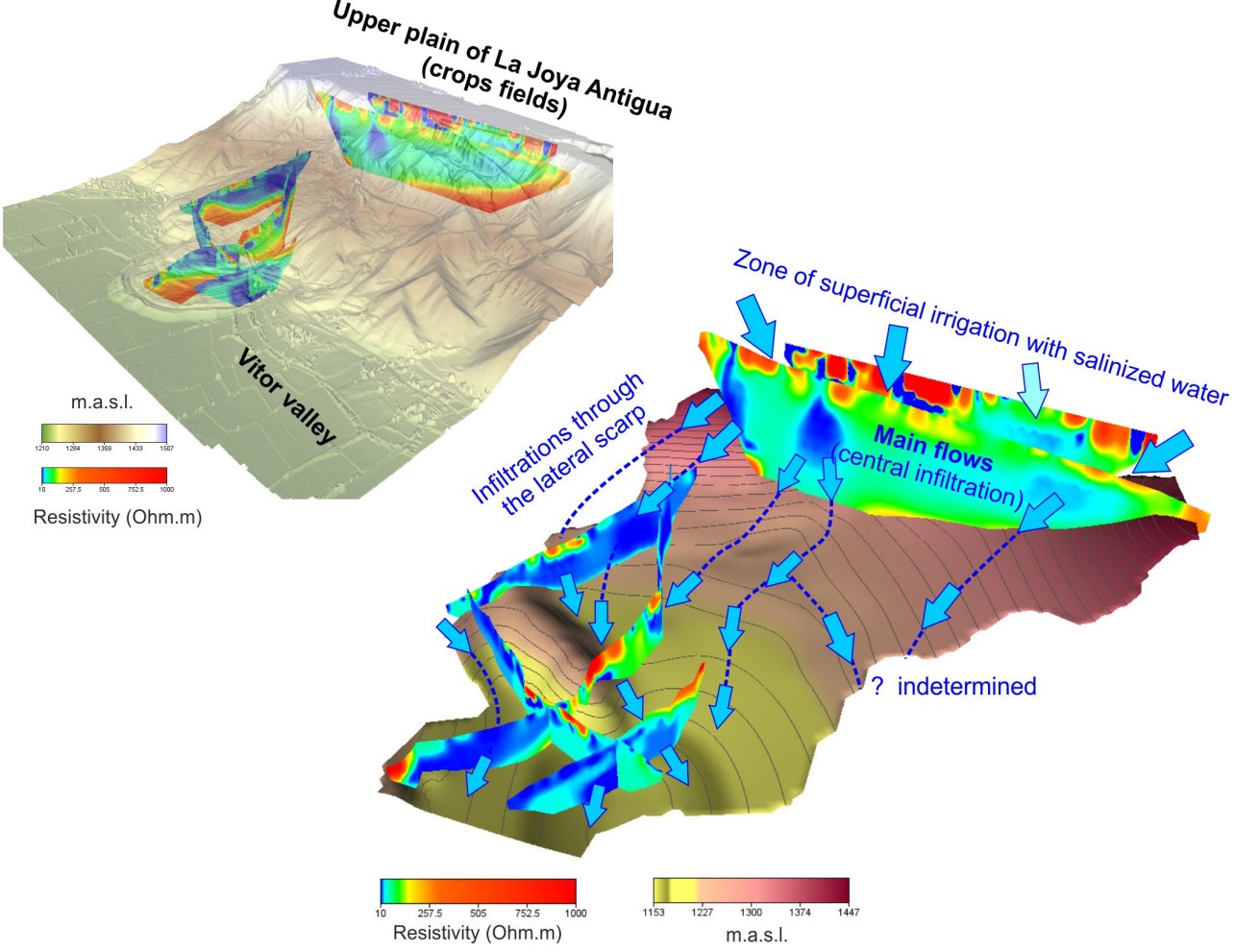

**Figure 10.** (**a**) A 3D layout of 2D-ERT electrical models below the digital elevation model. (**b**) Extrapolated top of basement roc (Lower Moquegua Fm.) from the 2D-ERT interpretation. The arrows indicate preferred flow directions.

The beginning of the SVP-02 seismic profile (Figure 9b) coincides with the end of the main scarp over the heat sector (Figures 3 and 6). In this place, the upper layer has a well-defined velocity of 1500 m/s, indicating that there is a significant inflow of water. The top basement rock is also clear (4000 m/s), and there is hardly any middle layer. Around a distance of 150 m, we can see an inner fault that would correspond structurally to the displacement of the minor escarp, characterized by a displacement of about 35 m (jump). If we correlated this seismic profile with the ERT-01 electric profile, the materials at the bottom of the rupture surface between 175 and 450 m long are moderately stiff and clear (3000 m/s), coinciding with medium-values resistivities.

*4.4. 3D Subsurface Models*

In order to obtain global knowledge of the Pie de Cuesta landslide, all geophysical profiles have been georeferenced and placed under the Digital Terrain Model (DTM) (Figure 10a). In this figure, we used only the ERT profiles to present a clear image. Once the profiles have been spatially distributed (GNSS georeferenced), the rupture surface has been deduced by digitizing the top of the resistive basement roc (Lower Moquegua Fm.) in each profile and extrapolating the point values for the landslide sector that has a reasonable geophysical coverage (Figure 10b). As described in the previous section (Figure 6), this digital and georeferenced rupture surface represents the subsurface where the sliding mass circulates and approximately coincides with the top of the strong roc.

Taking into account the fact that the upper plain corps (La Joya Antigua) is irrigated with salinized water that produces conductive effects, and considering the fact that when this water infiltrates, it follows along the maximum slopes of the rupture surface (at similar lithology), a first attempt has been made to establish the main flow entrances and their directions. Figure 10b shows the flow directions of water infiltrations; they are detected mainly in the sectors close to the lateral scarps and, apparently, with a greater water contribution on the north side, coinciding with the prominent toe of the subsurface.

Likewise, the 3D models in Figure 11 are the final results of the entire study. The two images above correspond to the DTM with a resolution of 2m/pixel (Figure 11a) and the deduced 3D rupture surface model without the electrical profiles (Figure 11b), giving 1,777,000 m$^2$ for a no-planar area and a 1,247,748 m$^2$ for a planar area. On this subsurface, the preferred runoff directions set out in Figure 10 have been placed. The model presented in Figure 11c is a composite that demonstrates the sliding subsurface related to the DTM, where each node of the mesh preserves its UTM coordinates; with the vertical (z) expressed in m above sea level (m.a.s.l., altitudes).

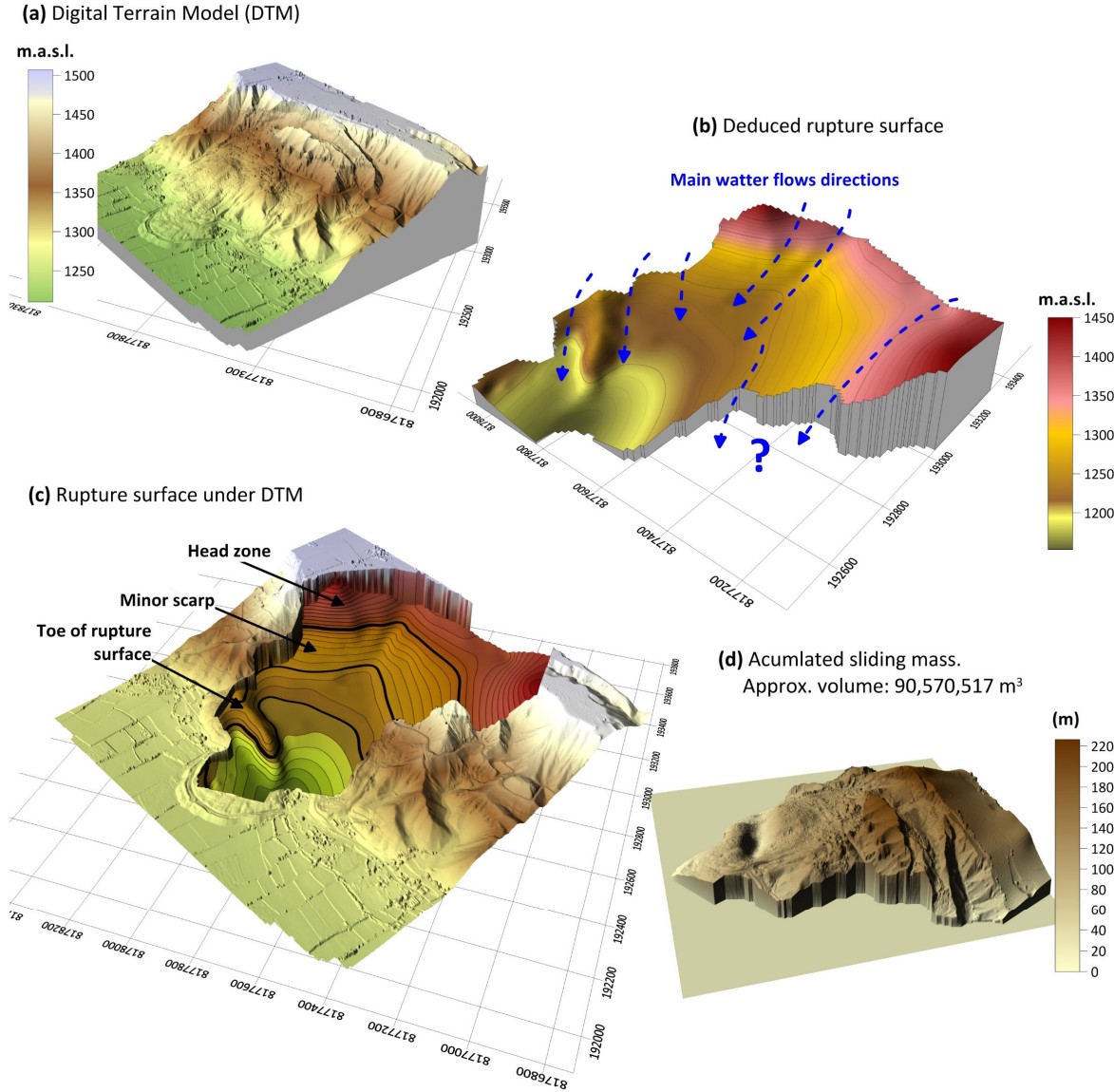

**Figure 11.** (**a**) Digital Terrain Elevation model (DTM) of the Pie de Cuesta landslide. (**b**) Deduced rupture surface from the geophysical methods. (**c**) Rupture surface under DTM. (**d**) Accumulated sliding mass.

If we compare this sliding subsurface with the graph of the inner geometry for the rotational/translational landslide in Figure 6, we can establish the main parts for the Pie de Cuesta slide: (i) the upper heat zone comprises altitudes between 1500 and 1300 m (first thick contour line), (ii) the minor scarp is placed at depths of 1300–1230 m (second thick contour line), and (iii) the toe surface rupture is the "relief" bordering on the north of the main lobe. It is noteworthy that the toe ends just behind the main lobe of the sliding and is not detected in the borderline profile ERT-04. This may be interpreted as a line of interception, in this case, capping between the bottom of the rupture surface and the original ground surface. Above it, the sliding mass flows with the fluidized materials. The seismic profile SRT-02 marks velocities of 500–1500 m/s for the involved materials.

Figure 11c demonstrates the difference between the MDT and the entire sliding sub-surface and represents the accumulated sliding mass along the major part of the landslide, giving a total volume of 90,570, 516 $m^3$ (about 90 $Hm^3$).

## 5. Discursion and Conclusions

The objective of this work has been to present the methodology followed in the study of a large landslide, as in the case of the active Pie de Cuesta slide. In all cases, a previous geological study is decisive in recognizing the lithologies present and understanding their behavior when they are saturated. In this case, the geological survey consisted of defining the geomorphology and the stratigraphic series and determining that it is mainly a rotational/translational type of landside. At present, this landslide is "almost inactive", but there is a high probability of reactivation because any small change in the water content or removal of the lower part can lead to a new great instability.

In this situation, the geophysical profiles that have been carried out provide valuable information on the internal geometry of the landslide. According to the results, the following conclusions can be drawn:

(i) With the electrical profiles, it has been possible to discriminate the sliding (conductive) mass from the resistive bedrock. Between these two layers, we identified the transition zone with medium resistivity values, which we related to a level of debris and the erosional top of the basement.

Another important contribution of electrical exploration has been that it is possible to establish, albeit only approximately, the preferred directions of water flow (by infiltrations or runoffs); according to the principle that for the same material, a decrease in resistivity can be associated with an increase in water content, especially when the impregnation water is highly mineralized, as in this case.

(ii) Although only two seismic profiles were made for the test, they have shown that when there is a good contrast between the densities of the materials involved, the obtained models provide good information on their compaction and degree of geotechnical safety.

When comparing the slip masses detected in seismic and electrical surveys (coincident), we can see that they coincide well when the top basement is a compact or aquitard formation. However, there are cases where the top basement is eroded, fractured, or partially dissolved, and the water infiltrates, reducing resistivities, while velocities are little affected (in the second order). Then, when the models disagree and only id electrical data are available, the indeterminacy arises. In the Pie de Cuesta landslide, there is the possibility of this happening in some sectors because the basement roc (Lower Moquegua Fm.) would have layers of gypsum on its top and below that would be very compact and dry clays. So, the levels of gypsum may be partially impregnated and saturated, and the resistivity decreases at a greater rate than the velocity, whereas in compacted dry clays, the velocities and resistivities remain high. So, the integration of all the data enabled us to better identify the geometry of the landslide.

With the 3D rupture surface model, it is possible to know the inner "landslide geometry", the main flow directions, and calculate the volume and distribution of the sliding mass. These results provide valuable information to gusset a geotechnical diagnostic. Having established this critical slip surface for a large area of the landslide gives an advantage

of the geophysical survey over borehole data, both for its high cost and for its very local information. However, in a general study, both techniques are crucial.

Finally, we consider this case study an interesting example of geophysical exploration on large landsides. The difficult topography caused by the multiple slumps and the thickness of the sliding mass present a significant handicap to placing the geophysical profiles. This is a crucial aspect because it greatly depends on the accessibility of the area and the availability of the physical space required. In our case, we need to extend profiles up to 1100 m long in order to obtain data at greater depths since this landslide is approximately 200 m tall.

Our last comment concerns the validity of the 3D geophysical model obtained. The most common way to verify the geophysical results is to check them using well logs. Although the boreholes only provide good information at one subsurface point, they are a useful tool for calibrating the geophysics profiles at their intersection point. This ensures the reliability of the rest of the profile. Unfortunately, no sounding has yet been carried out on this site to assess the degree of accuracy, and our geophysical survey has only been checked in the areas where a geological inspection of the "lateral-depths" could be carried out. mainly in lateral slopes and the final toe part.

**Author Contributions:** Conceptualization, J.S. and K.L.; methodology, T.T. and K.L.; software, Y.H. and K.L.; validation, T.T., K.L. and J.S.; resources, J.H., R.I. and Y.A.; writing—review and editing, Y.H., R.I., J.S., J.H., Y.A. and T.T.; supervision, J.S., K.L. and J.H.; All authors have read and agreed to the published version of the manuscript.

**Funding:** This research was funded by the Perú National Research Program funding the National University of San Agustín of Arequipa to carry out the Research Project grant number IBAIB-03-2018-UNSA. The APC was funded by the National University of San Agustín of Arequipa.

**Data Availability Statement:** Data are contained within the article.

**Acknowledgments:** Carlos Araque, for his important contribution in the development of 3D models for this research. To Fredy Perez, for his important support in field work and processing of seismic lines.

**Conflicts of Interest:** The authors declare no conflict of interest.

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
