# Peer review of "Using Electrical Resistivity Tomography Method to Determine the Inner 3D Geometry and the Main Runoff Directions of the Large Active Landslide of Pie de Cuesta in the Vítor Valley (Peru)"

_geosciences, doi:10.3390/geosciences13110342_

Round 1

Reviewer 1 Report

Comments and Suggestions for Authors

Dear authors!

In general, it is suitable for this publication. It would just be better to check the format of the references. For example, number 17 and 32. Not every reference has a DOI.

Line 16. “electrical” is probably missing after ERT.

Line 125. It would be nice to place a link to the description of the Elect Pro-10 device or indicate some its characteristics.

Figure 5. The pictures are too small. I suggest enlarging it.

Line 301, 314, 321. I suggest specifying the range of values in ascending order. That is, 10-20 instead of 20-10.

Line 462. Patents. Strange section title.

Author Response

The authors would like to thank you for your effort in reviewing our work. We have incorporated all your suggestions and, without a doubt, the manuscript has been improved.

SUMMARY:

“In general, it is suitable for this publication. It would just be better to check the format of the references. For example, number 17 and 32. Not every reference has a DOI.”

Sorry for these transcription errors. We have reviewed and corrected all references.

PARTICULAR COMMENTS:

Line 16. “electrical” is probably missing after ERT.

Of course!, thanks.  In the corrected manuscript we have put: ERT (electrical resistivity tomography)

Line 125. It would be nice to place de a link to the description of the Elect Pro-10 device or indicate some its characteristics.

According this comment we have incorporate the link of this resistivimeter and also some specifications.

Figure 5. The pictures are too small. I suggest enlarging it.

The images in Figure 5 have been enlarged.

Lines 301,314 and 321. I suggest specifying the range of values in ascending order. That is. 10-20 instead of 20-10.

Of course!, sorry. We have corrected the order of these values.

Line 462. Paten. Strange section title

This section figure in the “geociences_template.dot”  file. The specifications say that this is not a mandatory section but may be added if there are patents resulting from the work reported in this manuscript…..

GENERAL COMMENTS

QUESTION 1: The description of the used methods can be improved

According this comment we have further expanded and described the two methods used.

QUESTION 2: The results the results should be clearer

We have revised the explanation of the results by introducing further clarifications.

Reviewer 2 Report

Comments and Suggestions for Authors

The manuscript presented an very meaningful study on a method for distinguishing the inner structure of the landslide through the correlation between the resistivity models (2D-ERT), the Vp seismic velocties (2D-SVP) and the geological information.  The structure of the paper is clear and the language is readable. It can be accepted, and the following questions are suggested to address before publication. 

1. The basic principle of the electrical resistivity tomography should be introduced in this paper, and the specific research process and development history of this method in the field of geological survey, especially in the field of landslide geological body determination should be summarized in the introduction.

2. What are the advantages of the method introduced in this paper compared with the traditional geological survey method? Are there any deficiencies in precision or accuracy?

3. How can the results obtained by this method be verified?

Author Response

The authors would like to thank you for your effort in reviewing our work. We have incorporated all your suggestions and, without a doubt, the manuscript has been improved.

SUMMARY:

“The manuscript presented an very meaningful study on a method for distinguishing the inner structure of the landslide trough the correlation between the resistivity models (2D-ERT), the Vp seismic velocities (2D-Vp) and the geological information. The structure of the paper is clear and the language is readable. And the following questions are suggest to address before the publication. ”

SPECIFIC QUESTIONS:

QUESTION 1: The basic principle of the electrical resistivity should be introduced in this paper and the specific research process and development history of the methods in the field of geological survey, especially in the field of landslide geological body determination should be summarized in the introduction

According this comment we have further described the basics of the electrical c.c. method in the corresponding section and summarized geological applications in the Introduction Section.

QUESTION 2: What are the advantages of the method introduced in this paper compared with the traditional geological survey method? Are there any deficiencies in precision and accuracy?

As we say (not so clear) in this paper; any study of the terrain (in particular a landslide) it is always necessary to carry out a geological study as a first approach. But the advantage of using geophysical methods lies in the fact that they are non-destructive techniques of the subsurface exploration, from the surface. And the models obtained complement and explain the geological hypotheses and the cartographic maps of the study area. In this context, the aim of the two methods used in this work was to obtain an approximate model of the inner structure of the landslide and establish the preferred directions of the water flow by studying the electrical and seismic responses of the materials involved.

Thank you very much for this comment. In order to better introduce the objective of the study, we have incorporated it in the introduction.

QUESTION 3: How can the results obtained by this method be verified?

The most common way to verify the results obtained by geophysical methods is by mechanical soundings. Although the boreholes provide punctual information, they constitute an element of validation of the geophysical models that pass through the borehole.

Unfortunately, no sounding has yet been carried out on this site to assess the degree of accuracy of the geophysical results. 

We have again considered this question interesting and it has been incorporated in the conclusions.

Reviewer 3 Report

Comments and Suggestions for Authors

The manuscript presents a method to determine the inner 3D geometry  and the the preferred runoff directions of a landslide. The applied method is based on non-invasive electrical resistivity tomogrpahy. 

The manuscript topic mathches the journal scope and has a relevance in the field of geosciences. The authors applied their method to a given area and highlighted its benefits. I would recommend to highlight more how other researchers can benefit from your results.

The applied methodology as well as the previous geological survey were presented in a detailed way. The findings are relevant and supported by the research work. The style of the paper is of high quality.

Formatting:

The structure of the manuscript is well designed. The abstract gives a good sum up about the content of the paper. The figures, and tables help the reader to understand the research work and to implement the results.

Author Response

The authors would like to thank you for your effort in reviewing our work. We have incorporated all your comments  and, without a doubt, the manuscript has been improved.

SUMMARY:

“The manuscript presents a method to determine the inner 3D geometry and the preferred runoff directions of the landslide.  The applied method is based in non-invasive electrical resistivity tomography”.

GENERAL VALUATIONS:

“The manuscript topic matches the journal scope and has relevance in the fields of geosciences. The authors applied their method to a given area and highlighted its benefits. I would recommend to highlight more how other researchers can benefit from yours results”.

The authors would like to thank you for your positive assessment of the use of the electrical method for landslide studies. Although it is one of the most traditional geophysical methods of approach, we believe that the results presented have a certain originality that may be useful for future research on this topic.

“The applied methodologies as well as the previous geological survey were presented in a detailed way. The findings are relevant and supported the research work. The style of the paper is of high quality.”

This paper is part of a research project in which most of the participants (authors) are young researchers and the aim of the group was to make the work as honest as possible. Thank you very much.

FORMATTING:

“The structure of the manuscript is well designed. The abstract gives a good sum up about the content of the paper. The figures and the tables help the reader to understand the research work and to implement the results.”

Thank you again for this comment.  The graphic presentation has been one of the aspects they have taken care of. In particular, the resulting models should be self-explanatory (almost).

Reviewer 4 Report

Comments and Suggestions for Authors

geosciences-2659841-peer-review-v1

The manuscript “Use of non-invasive electrical resistivity tomography method to determine the inner 3D geometry and the preferred runoff directions of the large active landslide of Pie de Cuesta in the Vitor Valley, Arequipa-Peru” addresses an interesting topic, which adhere to Geosciences journal policies.

The manuscript tackles a topic of interest, related to landslide assessment in a region from Peru, by means of non-intrusive instrumentation and electric profiles. While the novelty of the manuscript is not very high, the manuscript presents an interesting case study, and can be considered for publication after several improvements.  

-       The main issue is that the manuscript feels more like a report, and less as a scientific paper. The novelty is not high, as it is a case study and evaluation of a region susceptible to slope failure due to water accumulation. My recommendation is to rewrite parts of the article, and focus more on the scientific value you bring in this article.

-       The abstract must be improved and must include more concrete results

-       Another main issue is the current structure of the article. In my opinion, most of the structure must be reconsidered. The introduction should be expanded. Subchapters 1.1 and 1.2 can be merged in order to have a Figure with the “Study area”. Also, this belongs more to the M&M chapter. The first part of Results chapter are not results. Those are general knowledge and Fig. 6 from Varnes is a well-known concept. In my opinion it should be moved to M&M

-       While the Methodology regarding electrical resistivity tomography is well presented, the data availability can be further expanded

-       Discussion chapter can be improved

-       The Citations are missing in the current manuscript

-       The overall English and formatting are lacking. I suggest moderate improvements  

The obtained figures are well constructed and displayed, and the interpretations are comprehensive. I congratulate the authors for the genuine work.

Additional articles worth taking into consideration for reference:

Sestras, P.; BilaÈ™co, Ș.; RoÈ™ca, S.; Veres, I.; Ilies, N.; Hysa, A.; Spalević, V.; Cîmpeanu, S.M. Multi-Instrumental Approach to Slope Failure Monitoring in a Landslide Susceptible Newly Built-Up Area: Topo-Geodetic Survey, UAV 3D Modelling and Ground-Penetrating Radar. Remote Sens. 2022, 14, 5822.

Comments on the Quality of English Language

I suggest moderate improvements  

Author Response

The authors would like to thank you for your effort in reviewing our work. We have incorporated all your suggestions and, without a doubt, the manuscript has been improved.

SUMMARY:

“The manuscript “Use of non-invasive electrical resistivity tomography method to determine the inner 3D geometry and the preferred runoff directions of the large active landslide of Pie de Cuesta in the Vitor Valley, Arequipa-Peru” addresses an interesting topic, which adhere to Geosciences journal policies.”

“The manuscript tackles a topic of interest, related to landslide assessment in a region form Peru, by means of non-intrusive instrumentation and electric profiles. While the novelty of the manuscript is not very high, the manuscript presents an interesting case study, and cab be considered for publication after several improvements.”

GENERAL COMMENTS

QUESTION 1: Moderate editing of English language and formatting are lacking.

Once the proposed corrections have been applied, we have proceeded to a revision of English language and the format.

QUESTION 2: The references can be improved

According this comment, we have incorporated more references. And we and we have considered the reference of Sestas et al. (2022) as a good example of a landslide 3D monitoring study in the domain of civil engineering. Reaching the 2D- GPR profile at 30 m depth.

But our landslide is a different case study: it has a huge size (1 km2) with a complex topography, where subsurface investigation to a depth of 200 m is required. In this context, our manuscript is a first approach in order to understanding the morphology of this landslide, i.e.: to know the internal structure and to detect the main flow directions. In addition, we have estimated the volume and distribution of sliding mass.

 For a later approach, accessible points will be chosen for borehole drilling in some placed where the geophysical crosses. With these punctual calibrations we will able to establish some geotechnical parameters.

We are grateful for this comment and have included it in the Conclusion Section.

QUESTION 3: The description of the used methods can be improved

According this comment we have further expanded and described the two methods used.

SPECIFIC COMMENTS:

QUESTION 4: The main issue is that the manuscript feels more like a report, and less as a scientific paper. The novelty is not high, as it is a case study and evaluation of a region susceptible to slope failure due to water accumulation. My recommendation is to rewrite parts of the article, and focus more on the scientific value you bring in this article.

This manuscript is obviously a case study of large landslides.  But it is publishable due to some novel aspects:

i) There is scares literature (or none?) presenting a methodology for obtaining a 3D model of the internal structure of a landslide that looks as good as the Vannes model does.

In this work, the top of the rupture surface has been digitised and extended from the geophysical profiles. In this way, a continuous 3D model has been obtained which has been introduced into the digital terrain model. Normally, in the literature only the authors marking this surface in the geophysical profiles and the rupture surface is presented as a "discontinuous model".

ii) There is scares literature (or none?) presenting a methodology for obtaining a 3D digital model of the internal structure of a landslide that looks as good as the Vannes model does.

Having a 3D digital model of the rupture surface means that calculations such as the volume of the sliding mass, the scarps slopes, etc. can be carried out. In addition, the spatial georeferenced distribution GNSS(x,y,x) of the electrical and seismic profiles has also provided the characterization of the preferred water flow directions. That it is valuable information in this type of landslides.

  •  The abstract must be improved and must include more concrete results

Of course!, sorry. We have introduced the concrete results

  • Another main issue is the current structure of the article. In my opinion, most of the structure must be reconsidered. The introduction should be expanded. Subchapters 1.1 and 1.2 can be merged in order to have a Figure with the “Study area”. Also, this belongs more to the M&M chapter. The first part of Results chapter are not results. Those are general knowledge and Fig. 6 from Varnes is a well-known concept. In my opinion it should be moved to M&M.

Thank you for this observation. We have seen the error ourselves. We have structured the text according to your suggestions.

We have considered your proposal to move Figure 6 to the M&M Section, or in the Study Area Section. But we believe that the Varnes scheme is more closely related to the interpretative basis introduced in Section 4.  Results and Interpretation.

  • While the Methodology regarding electrical resistivity tomography is well presented, the data availability can be further expanded.

According this comment we have incorporate some data specifications.

  • Discussion chapter can be improved

We have revised the explanation of this chapter and introducing further clarifications.

  • The obtained figures are well constructed and displayed, and the interpretations are comprehensive. I congratulate the authors for the genuine work.

The graphic presentation has been one of the aspects we have taken care of. In particular, the resulting models should be self-explanatory (almost).

This paper is part of a research project in which most of the participants (authors) are young researchers and the aim of the group was to make the work as honest as possible. Thank you very much for this comment.

Round 2

Reviewer 1 Report

Comments and Suggestions for Authors

Dear Ms. Huayllazo and co-authors!

All comments have been taken into account. The manuscript is ready for publication.

Reviewer 4 Report

Comments and Suggestions for Authors

The revised manuscript demonstrates the author’s commitment in improving the overall paper, thus obtaining a cohesive and interesting article. The authors gave comprehended responses to the concerns raised by the reviewers, and significantly improved their manuscript.

In my opinion, the manuscript can be considered for publication.